# Exercise Addiction and Intimate Partner Violence: The Role of Impulsivity, Self-Esteem, and Emotional Dependence

**DOI:** 10.3390/bs14050420

**Published:** 2024-05-17

**Authors:** Leticia Olave, Itziar Iruarrizaga, Marta Herrero, Patricia Macía, Janire Momeñe, Laura Macía, José Antonio Muñiz, Ana Estevez

**Affiliations:** 1Faculty of Health Sciences, International University of Valencia, 46002 Valencia, Spain; leticiamaria.olave@professor.universidadviu.com; 2Department of Experimental Psychology, Cognitive Processes and Speech Therapy, Faculty of Social Work, Complutense University of Madrid, 28223 Madrid, Spain; jomuniz@ucm.es; 3Department of Psychology, Faculty of Health Science, University of Deusto, 48007 Bilbao, Spain; m.herrero@deusto.es (M.H.); janiremomene@deusto.es (J.M.); lauramacia@deusto.es (L.M.); aestevez@deusto.es (A.E.); 4Department of Basic Psychological Processes and Their Development, University of the Basque Country, 20018 Donostia-San Sebastián, Spain; patricia.macia@ehu.eus

**Keywords:** exercise addiction, interpersonal partner violence, emotional dependence, impulsivity, self-esteem

## Abstract

Given the scarcity of studies linking exercise addiction to intimate partner violence, the present study aims to analyze the relationship between these variables and examine the potential mediating roles of emotional dependence, impulsivity, and self-esteem. This is a non-experimental, cross-sectional correlational design study. The sample comprised 887 university students (86% women, mean age 20.82 years, SD = 3.63). Elevated levels of exercise addiction were associated with increased impulsivity, emotional dependence, and exerted violence, as well as decreased self-esteem and perceived violence. Mediation models were tested, explaining 7% of the variance in received violence, 13% of the variance in exerted violence, and 6% of the variance in perceived violence. Higher levels of exercise addiction were linked to increased received and exerted violence and decreased perceived violence, attributed to the positive impact of exercise addiction on emotional dependence. This study highlights the mediating roles of self-esteem and impulsivity in the relationship between exercise addiction and partner violence. Identifying risk or vulnerability factors such as emotional dependence, impulsivity, and self-esteem related to exercise addiction and interpersonal partner violence is especially relevant for designing and implementing preventive interventions in the general young population.

## 1. Introduction

Behavioral addictions represent a serious social problem with significant repercussions in people’s lives, affecting their physical and mental health. Although major diagnostic classifications such as the Diagnostic and Statistical Manual of Mental Disorders (DSM-5) [1] or the International Classification of Diseases (ICD-11) [2] only recognize gambling disorder, gambling addiction disorder, and video game use disorder, the scientific community is concerned about this phenomenon. Increasing research is being conducted on other behavioral addictions, including exercise addiction (EA) [3,4,5]. EA involves dysfunctional behavior characterized by compulsive physical training [6], despite clear negative health consequences and a significant loss of control over behavior [7,8,9]. It falls within the obsessive–compulsive and impulsive spectrum of behavioral addictions [6] and is defined by elements traditionally associated with substance addictions, such as withdrawal, tolerance, desired effects, euphoria, interpersonal conflicts, time spent on behavior, reduced time spent on other important activities, persistence of behavior despite negative consequences, loss of control, and relapses [8,10,11,12,13].

However, it is important to note the duality regarding physical exercise, as incorporating regular physical activity into daily routines entails a series of significant health and well-being benefits for individuals. This practice, recommended by the World Health Organization [14], not only strengthens the body physically but also benefits mental and emotional health. From a biopsychosocial perspective, regular exercise helps maintain a healthy weight, reduces the risk of chronic diseases such as diabetes and heart disease, and improves mood by releasing endorphins and reducing stress. Additionally, it fosters social interaction by participating in group activities and promotes an overall sense of well-being and quality of life. Individuals who do not engage in sufficient physical activity face a 20% to 30% increase in the risk of mortality compared to those who maintain an adequate level of physical activity [15]. Despite the multiple positive effects of physical exercise, when physical exercise is practiced excessively and significantly interferes with an individual’s daily life, it can lead to unhealthy behavior rather than contributing to health status.

In the etiology of such addictions, impulsivity is of particular relevance [16,17,18,19,20,21,22]. According to the behavioral disinhibition model postulated by the Center for Education and Drug Abuse Research (CEDAR) team founded by the National Institute of Drug Abuse (NIDA), decreased prefrontal cortex response is related to impulsivity problems and the inhibitory control system, increasing the risk of addiction [23,24].

Similarly, both substance addictions (such as alcohol, tobacco, marijuana, and other drugs) and behavioral addictions (EA, addiction to social networks and smartphones, emotional dependence, and gambling disorder, among others) are associated with low self-esteem, being a significant risk factor in the initiation and maintenance of addiction [25,26,27,28,29,30,31,32,33]. Moreover, impulse control issues and low self-esteem are closely related to other problems, such as emotional dependence and interpersonal partner violence [19,34].

Interpersonal partner violence poses a significant challenge in societies, with a worrying trend given its incidence and profound impact on the psychological and physical health of the victims [35,36]. This form of interpersonal violence tends to increase in frequency and intensity as the relationship progresses [37], manifesting in three main forms: psychological, physical, and sexual [38,39,40]. Of them, psychological violence is the most prevalent [41]. Previous research has shown a positive association between partner violence and emotional dependence, as well as various types of addictions, negatively impacting violent dynamics and facilitating the maintenance of conflictive relationships [33,42,43].

Emotional dependence is fundamentally characterized by an intense need for affection towards a partner [44] due to unmet emotional needs that are attempted to be alleviated in a maladaptive manner through interpersonal relationships [32]. The role of emotional dependence is even more relevant given its association with the persistence of a partner in a violent relationship [18,33,42,45,46]. In this regard, emotional regulation plays a fundamental role, acting as a predictive factor alongside psychological abuse in emotional dependence [47].

Therefore, the question arises as to whether self-esteem, impulsivity, and emotional dependence influence the relationship between exercise addiction and intimate partner violence. This is posed because emotional dependence plays a role in the persistence of romantic relationships, and in some studies, it acts as a mediator between certain psychological variables and violence [33,42,43,48]. Although there is evidence of a relationship between emotional dependence and addictions, research on emotional dependence and exercise addiction has not yet been conducted [49]. Similarly, low self-esteem and high impulsivity have been implicated in violent relationships and are, in turn, related to emotional dependence. Therefore, the proposed hypothesis is that impulsivity and self-esteem will serve as the first parallel mediators, while emotional dependence will serve as a second mediator in the relationship between exercise addiction and intimate partner violence.

Despite this significant issue, there are no previous studies linking exercise addiction to partner violence. Therefore, the aim of this study is to analyze the relationship between EA and interpersonal partner violence and observe whether impulsivity, self-esteem, and emotional dependence act as mediating variables between them.

## 2. Materials and Methods

This study adopts a cross-sectional, correlational, non-experimental design. The sample was incidentally drawn from students attending the Complutense University of Madrid, utilizing snowball sampling to gather participants. Data collection was conducted through pencil and paper surveys administered in university classrooms. Prior to participating, all individuals provided informed consent. Furthermore, this study adhered to the principles outlined in the Declaration of Helsinki [50].

This study was conducted in strict adherence to the ethical principles applicable to psychological research. Approval was obtained from the Deontological Commission of the Faculty of Psychology at the Complutense University of Madrid (UCM), thus demonstrating a commitment to respecting the deontological standards established by the Official College of Psychologists and the Scientific Societies of Psychology. Our study was conducted under the reference number 2020/21-035, ensuring that all ethical measures and procedures were rigorously followed to guarantee the integrity and well-being of the participants involved.

### 2.1. Participants

The participants in this study were 887 university students whose age was 20.82 years old (DT = 3.63), of whom 86% were women and 14% were men.

Table 1 presents the data on exercise practice and diet. More than half of the sample described practicing physical exercise (61.3%). Slightly less than a third of the sample (29.7%) described regularly attending a gym for a mean of 3.33 days per week (SD = 1.22) during an average of 1.56 h per day (SD = 0.58). The main reason for exercise practice at the gym was to be healthy and fit. Approximately a third of the sample (35.5%) described practicing exercise outside the gym 3.05 days per week (SD = 1.48). The mean number of hours of practice outside the gym was 1.61 h per day (SD = 1.00). The most frequent reason for this practice was also to be healthy and fit. A total of 11% followed a diet, with the most common being vegetarian, followed by a hypocaloric diet.

### 2.2. Measures

#### 2.2.1. Exercise Addiction

Exercise addiction was measured using the Exercise addiction inventory [8]. The Spanish adaptation was conducted by [51], and it was validated among college students. It is a six-item self-reported survey with a 5-point Likert-type scale response format (1 = I completely disagree; 5 = I totally agree). It establishes cutoff points to create classifications with respect to the risk of presenting exercise addiction or excessive exercise; these are “no symptoms”, ”symptoms of exercise addiction”, and “risk of exercise addiction”. A global indicator of exercise addiction was computed for this study as the summation of the item scores, so higher levels of this variable indicated higher exercise addiction. In the present study, the level of internal consistency, measured through Cronbach’s alpha, was 0.83 (α = 0.83).

#### 2.2.2. Self-Esteem

We used the Rosenberg Self-Esteem Scale [52] to measure self-esteem. It consists of 10 items and a 4-option Likert-type response scale with 5 response options from 1 (“strongly disagree”) to 4 (“strongly agree”). It is widely used internationally and has been validated in the Spanish population by [53], as well as among college students [54]. The summation of the items was used as an indicator of the variable, so higher levels indicated higher self-esteem. It contains adequate psychometric properties, showing a Cronbach’s alpha of 0.89 in the present study (α = 0.89).

#### 2.2.3. Impulsivity

Impulsivity was measured using the BARRAT-BIS 11 impulsivity scale [55]. In this work, the Spanish adapted version of [56] is used, with this adaptation being measured in a university population. This is a self-report instrument that assesses impulsivity. It consists of 30 items that form a total impulsivity scale, which in turn is divided into 4 subscales: (a) cognitive impulsivity; (b) motor impulsivity; (c) unplanned impulsivity; and (d) total impulsivity. In this research, only the total impulsivity scale was used. It is answered by means of a Likert-type scale with 4 response options from 0 (“rarely or never”) to 4 (“always or almost always”). The global dimension of impulsivity was used in this study as a summation of the items, so higher levels of the variable were indicators of higher impulsivity. It has acceptable internal consistency values; the Cronbach’s alpha value for the total scale in this study was 0.76 (α = 0.76).

#### 2.2.4. Emotional Dependence

We used the Emotional Dependence in Dating (DEN) scale [57] to measure emotional dependence. This questionnaire has been validated in Spanish university youth and consists of 12 items gauging emotional dependency by measuring the frequency of feelings such as (a) the need to avoid being alone, encompassing actions taken to evade being alone; (b) the need for exclusivity, indicating the need for their partner’s exclusive availability; (c) the need to please, i.e., actions taken to satisfy another individual while disregarding one’s own needs; and (d) the development of an asymmetrical relationship, reflecting the subordinate nature of the relationship. The instrument also provides a total emotional dependence scale. Responses are recorded on a 6-point Likert scale ranging from 0 (never) to 5 (always). The global indicator of emotional dependence was used in this study. It was calculated as the sum of all the scale items, so higher scores indicated higher emotional dependence. In terms of internal consistency, the alpha coefficient was 0.84 (α = 0.84). In this study, only the total emotional dependence scale was used.

#### 2.2.5. Partner Violence

Partner violence was measured with the Received, Exercised, and Perceived Violence in Young People’s Dating Relationships test [58]. This test has also been validated in Spanish youth from diverse academic backgrounds, including university settings. This is a self-report questionnaire consisting of 28 violence-related scenarios, measuring violence experienced by the informant in their romantic relationships; violence perpetrated by the participants against their partners; and perception of violence in the behaviors identified as experienced or exercised. Exercised or received violence is assessed on a 6-point Likert scale (ranging from “0 = never” to “5 = more than 15 times”). Meanwhile, the perception of violence offers 5 response alternatives: not violent, slightly violent, somewhat violent, fairly violent, and very violent. Participants were required to have been in a relationship for at least one month to complete the questionnaire. The violence scenarios are categorized into 5 types: physical violence (any non-accidental act causing or likely to cause bodily harm), sexual violence (imposition of contact and certain sexual practices against the will of the victim), psychological–social violence (behaviors involving isolation and avoidance of contact with others), psychological–humiliation (acts of ridicule, humiliation, verbal threats and insults, or threats of abandonment if the other person’s demands are not met), and psychological–control (behaviors involving control and jealousy). In the present study, the scores of received, exercised, and perceived violence were computed as the mean of the item scores following the original questionnaire. Higher scores indicated higher violence. The subscales obtained an adequate Cronbach’s alpha coefficient, as follows: exerted violence (α = 0.90), received violence (α = 0.96), and perceived violence (α = 0.97).

### 2.3. Procedure

During scheduled class sessions, students completed the questionnaires in their classrooms. They were assured that participation was voluntary, and if they chose not to participate, they could return a blank questionnaire without repercussion. Additionally, students were assured that their responses would remain confidential and anonymous, utilized solely for statistical analysis. No compensation was provided to the students for their participation. They were allotted between 20 and 30 min to complete the questionnaires. Informed consent was obtained from all subjects involved in this study.

### 2.4. Analytical Procedure

Analyses were performed with SPSS v27.0 [59]. First, the descriptive statistics and the correlations between the variables of our study (i.e., exercise addiction, self-esteem, impulsivity, and partner violence) were carried out. Second, the hypothesized models were tested with custom models in PROCESS v4.0 [60]. Three models were computed, one per dependent variable (i.e., received violence, exerted violence, and perceived violence). In all models, exercise addiction was included as an independent variable, self-esteem and impulsivity as the first parallel mediators, and emotional dependence as the second mediator (see Figure 1). Age and gender were included as control variables in the models. Additionally, to reduce the bias in the estimation of the indirect effects [60], 10,000 bootstrap samples were applied in the computation of the indirect effects and the corresponding 95% confidence intervals.

## 3. Results

First, the descriptive statistics and correlations are shown in Table 2. Exercise addiction was significantly correlated to all variables but received violence. Concretely, higher levels of exercise addiction were related to higher impulsivity, emotional dependence, and exerted violence, as well as lower levels of self-esteem and perceived violence. Self-esteem and impulsivity were negatively correlated. Both variables were significantly correlated to emotional dependence and received and exerted violence (self-esteem positively correlated and impulsivity negatively correlated), but they were not significantly correlated to perceived violence. Emotional dependence was positively correlated to perceived and exerted violence and negatively correlated to perceived violence.

Second, the mediation models were tested. These models explained 7% of the variance of received violence, 13% of the variance of exerted violence, and 6% of the variance of perceived violence. As displayed in Table 3, in all models, exercise addiction was related to lower self-esteem, higher impulsivity, and higher emotional dependence. Additionally, higher self-esteem was related to lower emotional dependence, but impulsivity was not related to the latter.

Regarding received and exerted violence, impulsivity and emotional dependence had positive direct effects on both indicators of partner violence, but the direct effects of self-esteem and exercise addiction were not significant. Thus, higher impulsivity and higher emotional dependence were related to higher received and exerted violence.

Regarding perceived violence, exercise addiction and emotional dependence had significant negative effects on this type of partner violence, but neither self-esteem nor impulsivity did. Consequently, higher exercise addiction and emotional dependence were directly related to lower perceived violence.

The indirect effects are displayed in Table 4. Emotional dependence mediated, by itself, the indirect effect of exercise addiction on received, exerted, and perceived violence. Thus, higher exercise addiction levels were related to higher received and exerted violence and to lower perceived violence due to the positive effect of exercise addiction on emotional dependence.

Self-esteem did not significantly mediate the effect of exercise addiction on any of the indicators of partner violence by itself. However, self-esteem significantly mediated the effect of the three types of partner violence through its effect on emotional dependence. Concretely, a higher risk of exercise addiction was related to higher received and exerted violence and to lower perceived violence through the effect on self-esteem and the effect of self-esteem on emotional dependence.

Furthermore, the results of the indirect effects showed that impulsivity mediated the effect of exercise addiction on received and exerted violence but not on perceived violence. Thus, higher exercise addiction led to higher received and exerted violence through its positive effect on impulsivity.

## 4. Discussion

The aim of this study was to analyze the relationship between EA and interpersonal partner violence and observe whether impulsivity, self-esteem, and emotional dependence act as mediating variables between them.

The first objective was aimed at analyzing the correlations between exercise addiction, self-esteem, impulsivity, emotional dependence, and partner violence (received, perpetrated, and perceived). The results revealed that as exercise addiction (EA) increased, so did impulsivity, emotional dependence, and perpetrated violence, while self-esteem and perception of violence decreased. This suggests that individuals with high levels of EA may have difficulties regulating their emotions and behaviors, acting impulsively in different situations. In this regard, as [14] indicate, the role of addictions and impulsivity mutually reinforce each other, as an impulse control deficit facilitates addiction onset, while the brain mechanisms that activate addictions also promote impulsive behavior. This is consistent with [61]’s model, which explains that at the root of addiction lies a disorder affecting impulse control and manifesting compulsively. The authors of [62] found similar results, indicating that poor emotional regulation and impulsivity are related to behavioral addictions such as gambling disorder and video game abuse, as well as greater dysfunctional psychological symptomatology. Likewise, the positive relationship between EA and emotional dependence suggests that individuals with high EA may seek approval and support from others to compensate for feelings of emptiness and existential uncertainty, facilitated by low self-esteem. Similar results have been found previously, where a relationship between emotional dependence and exercise addiction mediated by dysfunctional attachment styles has been observed [63]. Furthermore, the results reveal that emotional dependence is closely related to violence, as higher emotional dependence is associated with more perpetrated and received violence and less perceived violence. In this sense, it should be noted that previous studies warn that emotional dependence increases as psychological abuse increases [64,65], making violence perception difficult. The role of self-esteem has also been depicted in previous works, finding a correlation between low self-esteem and other behavioral addictions [66,67], including exercise addiction [68,69,70]. The importance of self-esteem is broader, as better self-esteem seems to correlate with lower impulsivity, suggesting it as a factor to consider. Additionally, it has also been found that higher EA is associated with more perpetrated violence, so EA could be related to feelings of anger, frustration, and increased aggressiveness, which can lead to violent behavior towards others. This tendency towards aggressiveness and anger, high impulsivity and emotional dependence, and low self-esteem, along with difficulty in regulating behavior, would increase the risk of violence towards the partner.

Secondly, the mediation models used in this study revealed that exercise addiction has a significant impact on various aspects of violence in intimate relationships. It was found that this addiction was related to a decrease in self-esteem, an increase in impulsivity, and greater emotional dependence. These findings align with previous research suggesting that certain addictive behaviors, such as exercise addiction, may be associated with underlying psychological issues, such as low self-esteem and a tendency towards impulsivity [70,71,72,73,74].

Regarding received and perpetrated violence, the results highlight the importance of modulating variables. Although there is no direct effect of EA on perpetrated and received violence, there is an indirect effect through impulsivity and emotional dependence, emphasizing the importance of these variables as risk or protective factors to be considered. This implies that individuals with higher levels of impulsivity or emotional dependence are more likely to experience or perpetrate violence in their relationships. These results are consistent with previous research that has identified impulsivity as a risk factor for violence in intimate relationships [75,76,77,78]. On the other hand, regarding the perception of violence, the results suggest that individuals with high levels of exercise addiction or emotional dependence may perceive less violence in their relationships, possibly due to a distortion in the perception of interpersonal interactions. It is also important to highlight that, although self-esteem did not directly mediate the effect of exercise addiction on violence in intimate relationships, it was found to mediate this effect through its relationship with emotional dependence. These results emphasize the importance of self-esteem in the dynamics of intimate relationships, particularly in how it influences emotional dependence, which in turn can affect the perception and perpetration of violence.

Finally, it was discovered that impulsivity mediated the effect of exercise addiction on received and perpetrated violence but not on perceived violence. This suggests that impulsivity may be a more relevant factor in the direct manifestation of physical or emotional violence in intimate relationships, while the perception of violence may be more influenced by other factors, such as emotional dependence. These results coincide with previous research that has identified impulsivity as a key predictor of violence in intimate relationships and positions emotional dependence as a key factor for remaining in violent relationships [42,43,44,45,46].

This study is not exempt from limitations. The first one refers to the characteristics of the study design since, being a non-experimental cross-sectional correlational design, causal relationships between the analyzed variables cannot be established through the obtained results. Additional constraints exist regarding the assessment instruments, as they rely on self-reported measures and serve as screening tools without diagnostic validity [79]. Moreover, the statistical analyses in this study were conducted within the general population, prompting the next research objective to investigate these variables among individuals with EA and ascertain any distinctions from control participants. Lastly, due to the sample being mainly composed of female university students (86%), the applicability of the findings to the general population is limited.

## 5. Conclusions

The conclusions of this study offer an innovative perspective on the relationship between exercise addiction (EA) and partner violence, highlighting the mediating role of psychological variables such as impulsivity, self-esteem, and, especially, emotional dependence. The novelty of this work is emphasized due to the scarcity of previous studies that relate exercise addiction to emotional dependence and even less so to violence in romantic relationships.

The connection between EA and emotional dependence, as well as their association with partner violence, underscores the importance of addressing addictive behaviors and emotional regulation issues in interventions aimed at preventing gender-based violence. Specifically, it is observed that impulsivity and emotional dependence act as risk or protective factors in received and exercised violence, highlighting the need to consider these aspects in intervention programs.

Lastly, it is highlighted that impulsivity mediates the effect of exercise addiction on received and exercised violence, while the perception of violence may be influenced by other factors, such as emotional dependence. These findings underscore the complexity of the mechanisms underlying partner violence and emphasize the importance of addressing multiple psychological variables in understanding this phenomenon.

Taken together, these findings provide new perspectives on addressing the relationship between exercise addiction and gender-based violence, emphasizing the need for comprehensive interventions that address both behavioral and emotional aspects of this issue.

## Figures and Tables

**Figure 1 behavsci-14-00420-f001:**
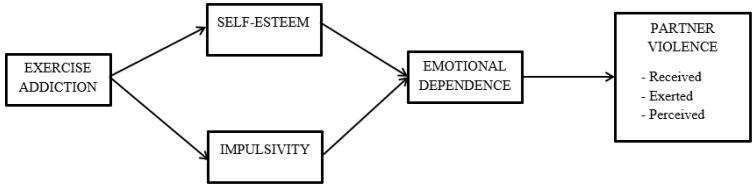
Hypothesized models.

**Table 1 behavsci-14-00420-t001:** Exercise and diet information for the sample.

Variable		n	%
Gym reasons			
	Being healthy/fit	187	71.9
	Aesthetic	20	7.7
	Self-esteem	4	1.5
	Socialization	14	5.4
	Gaining weight	0	0.0
	Losing weight	21	8.1
	Profession/competition	3	1.2
	Relax/leisure	11	4.2
Outside gym reasons			
	Being healthy/fit	234	72.0
	Aesthetic	0	0.0
	Self-esteem	0	0.0
	Socialization	0	0.0
	Gaining weight	2	0.6
	Losing weight	26	8.0
	Profession/competition	11	3.4
	Relax/leisure	52	16.0
Type of diet			
	Hypocaloric	30	29.1
	Hypercaloric	12	11.7
	Celiac	5	4.9
	Vegetarian	49	47.6
	Vegan	2	1.9
	No lactose	2	1.9
	Diabetic	3	2.9

Note: Some questions contained missing data.

**Table 2 behavsci-14-00420-t002:** Descriptive statistics and correlations of the continuous study variables (n = 887).

Variable	M	SD	1	2	3	4	5	6
1. Exercise addiction	10.73	5.11						
2. Self-esteem	32.10	5.77	−0.10 **					
3. Impulsivity	66.53	9.26	0.08 *	−0.21 ***				
4. Emotional dependence	10.56	8.27	0.17 *	−0.31 ***	0.13 ***			
5. Received violence	0.40	0.73	0.06	−0.13 ***	0.11 **	0.23 **		
6. Exerted violence	0.20	0.33	0.09 **	−0.08 *	0.15 ***	0.31 ***	0.57 ***	
7. Perceived violence	4.24	0.79	−0.12 ***	0.04	<0.01	−0.22 ***	−0.10 **	−0.26 ***

Note: The scores of variables 1–4 were computed as summations previous to the mean computation following the original instruments. * *p* < 0.05; *** p* < 0.01; *** *p* < 0.001.

**Table 3 behavsci-14-00420-t003:** Regression direct effects of the models.

	Dependent Variable					
Independent Variable	Self-Esteem	Impulsivity	Emotional Dependence	Received Violence	Exerted Violence	Perceived Violence
Exercise addiction	−2.89 **	2.24 *	3.75 ***	0.68	1.25	−2.34 *
Self-esteem			−8.81 ***	−1.64	1.10	−0.57
Impulsivity			1.72	2.33 *	3.70 ***	0.80
Emotional dependence				5.90 ***	9.45 ***	−5.95 ***

Note: * *p* < 0.05; ** *p* < 0.01; *** *p* < 0.001.

**Table 4 behavsci-14-00420-t004:** Indirect effects of exercise addiction on partner violence.

	Dependent Variable		
	Received Violence	Exerted Violence	Perceived Violence
Mediator	Ind [95% CI]	Ind [95% CI]	Ind [95% CI]
Self-esteem	0.0008 [−0.0002, 0.0024]	−0.0002 [−0.0007, 0.0001]	0.0003 [−0.0008, 0.0015]
Impulsivity	0.0008 * [0.0001, 0.0020]	0.006 * [0.0001, 0.0012]	0.0003 [−0.0005, 0.0014]
Emotional dependence	0.0005 * [0.0001, 0.0020]	0.0025 * [0.0011, 0.0041]	−0.0039 * [−0.0069, −0.0016]
Self-esteem ⟶ Emotional dependence	0.0012 * [0.0014, 0.0016]	0.0006 * [0.0002, 0.0011]	−0.0009 * [−0.0017, −0.0002]
Impulsivity ⟶ Emotional dependence	0.0001 [<0.0001, 0.0004]	0.0001 [<0.0001, 0.0003]	−0.0001 [−0.0004, <0.0001]

Note: Ind [95% CI] = indirect effect [bootstrap 95% CI]. * Significant indirect effects based on bootstrap 95% CI.

## Data Availability

Dataset available upon request from the authors.

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
