# Peer review of "Exercise Addiction and Intimate Partner Violence: The Role of Impulsivity, Self-Esteem, and Emotional Dependence"

_behavsci, 2024, doi:10.3390/bs14050420_

Round 1

Reviewer 1 Report

Comments and Suggestions for Authors

Exercise Addiction and Intimate Partner Violence: The Role of Impulsivity, Self-Esteem, and Emotional Dependence

I would like to thank the editor and authors for the opportunity to review this manuscript.

The topic of the study mainly focuses on the relationship between these variables and examine the potential mediating roles of emotional dependence, impulsivity, and self-esteem. The writing style is clear and the content well organized. However, there are a number of issues that the authors need to take into consideration:

Materials and Methods

--Lines 101 - I believe that the form in Table 1 needs to be reorganized. Specifically, the length of the table will need to be reduced. It is necessary to refer to the format of Table 2.

--Lines 103 145 - There are several corrections to this. First, it is necessary to rewrite the detailed measurement tool description in accordance with the basic format of the paper. Here, the standard refers to the usual quantitative research writing. Second, I think that the figures about the results of factor analysis (including sub-variables) should be presented in a table.

References

- Make sure to follow the journal’s reference guidelines.

Comments on the Quality of English Language

Exercise Addiction and Intimate Partner Violence: The Role of Impulsivity, Self-Esteem, and Emotional Dependence

I would like to thank the editor and authors for the opportunity to review this manuscript.

The topic of the study mainly focuses on the relationship between these variables and examine the potential mediating roles of emotional dependence, impulsivity, and self-esteem. The writing style is clear and the content well organized. However, there are a number of issues that the authors need to take into consideration:

Materials and Methods

--Lines 101 - I believe that the form in Table 1 needs to be reorganized. Specifically, the length of the table will need to be reduced. It is necessary to refer to the format of Table 2.

--Lines 103 145 - There are several corrections to this. First, it is necessary to rewrite the detailed measurement tool description in accordance with the basic format of the paper. Here, the standard refers to the usual quantitative research writing. Second, I think that the figures about the results of factor analysis (including sub-variables) should be presented in a table.

References

- Make sure to follow the journal’s reference guidelines.

Author Response

Dear reviewer 1

Thank you very much for the review conducted. All your comments, suggestions, and corrections have been taken into account and applied. Below, we respond to your kind comments.

--Lines 101 - I believe that the form in Table 1 needs to be reorganized. Specifically, the length of the table will need to be reduced. It is necessary to refer to the format of Table 2.

We have reduced the length of Table 1 by stating the percentage of some variables in text and reduced the width of Table 2.

--Lines 103 – 145 - There are several corrections to this. First, it is necessary to rewrite the detailed measurement tool description in accordance with the basic format of the paper. Here, the standard refers to the usual quantitative research writing. Second, I think that the figures about the results of factor analysis (including sub-variables) should be presented in a table.

Following Reviewer 1 suggestions, Figures 2 and 3 have been replaced with Table 3 (previous Table 3 is now labelled as Table 4). We have also rewritten the detailed measurement tool description in accordance with the basic format of the paper (taking as reference the following work published in this same journal. Li, M., Liu, F., & Yang, C. (2024). Teachers’ Emotional Intelligence and Organizational Commitment: A Moderated Mediation Model of Teachers’ Psychological Well-Being and Principal Transformational Leadership. Behavioral Sciences, 14(4), 345.)."

--Make sure to follow the journal’s reference guidelines.

 We have reviewed the references and adjusted the entire list to APA 7 following the instructions of the web site “Your references may be in any style, provided that you use the consistent formatting throughout”.

Thanks in advance for your time

Best regards

Itziar Iruarrizaga

Reviewer 2 Report

Comments and Suggestions for Authors

It would be an honor to review your work. This reviewer proposes the following comments.

First, the introduction only emphasizes the negative aspects of exercise addiction. Exercise addiction is academically reported to have two sides. Even though it has positive aspects, emphasizing only the negative aspects can cause misunderstanding in readers. Accordingly, the duality of exercise addiction needs to be further described.

In addition, there is a need to supplement the logical basis for each variable needed to establish a research hypothesis. Additional descriptions supplementing the parts described by the researcher with examples and references are required.

Research method

It is necessary to describe in detail how the average value of the survey tool was calculated. There is a significant difference in the range of the average values of each variable presented in <table 2>. If a 5-point Likert scale is used, the average value must be 5 or less, and the score calculation method for each factor must be described in detail.

Author Response

Dear reviewer 2

Thank you very much for the review conducted. All your comments, suggestions, and corrections have been taken into account and applied. Below, we respond to your kind comments.

--First, the introduction only emphasizes the negative aspects of exercise addiction. Exercise addiction is academically reported to have two sides. Even though it has positive aspects, emphasizing only the negative aspects can cause misunderstanding in readers. Accordingly, the duality of exercise addiction needs to be further described.

In the introduction, a description has been added about the duality of physical exercise practice, referring to the positive effects it has on people's health and its importance from the biopsychosocial model perspective (references have been added).

--In addition, there is a need to supplement the logical basis for each variable needed to establish a research hypothesis. Additional descriptions supplementing the parts described by the researcher with examples and references are required.

A paragraph has been added at the end of the introduction, before explaining the objectives, in which the relationship between the proposed variables (references have been added) and the hypothesis we pose is explained.

 --It is necessary to describe in detail how the average value of the survey tool was calculated. There is a significant difference in the range of the average values of each variable presented in <table 2>. If a 5-point Likert scale is used, the average value must be 5 or less, and the score calculation method for each factor must be described in detail.

The variables of exercise addiction, self-esteem, impulsivity and emotional dependence were computed as summation of the corresponding item scores previous to the sample mean computation following the original scales. Further explanation was included in the instrument section and in the Note of Table 2.

Thanks in advance for your time

Best regards

Itziar Iruarrizaga

Reviewer 3 Report

Comments and Suggestions for Authors

This study focused on exercise addiction and its association with intimate partner violence which is a subject that has not been extensively explored in academic research. A non-experimental, cross-sectional correlational design is applied. The study offered a deeper understanding of the relationships between exercise addiction and intimate partner violence. Its findings provided potential information for the future research in this field. Although age and gender were included as control variables, it is still a concern that 86% of participants are women.

Author Response

Dear reviewer 3

Thank you very much for the review conducted. All your comments, suggestions, and corrections have been taken into account and applied. Below, we respond to your kind comments.

-- This study focused on exercise addiction and its association with intimate partner violence which is a subject that has not been extensively explored in academic research. A non-experimental, cross-sectional correlational design is applied. The study offered a deeper understanding of the relationships between exercise addiction and intimate partner violence. Its findings provided potential information for the future research in this field. Although age and gender were included as control variables, it is still a concern that 86% of participants are women.

Taking into account the aspect kindly pointed out by reviewer 3, a more detailed explanation has been included in the study's limitations (references have been added).

Thanks in advance for your time

Best regards

Itziar Iruarrizaga

Round 2

Reviewer 1 Report

Comments and Suggestions for Authors

I think the authors have been working very hard to revise this paper, but I am going to suggest comment on a few minor points.

First, the widths of Table 1 and Table 2 still do not match.

Second, information on whether the measurement tools are suitable for university students should be presented in detail. For example,

Line 144-145,

"Impulsivity was measured using the BARRAT-BIS 11 impulsivity scale [55]. In this work, the Spanish adapted version of [56] is used."

The tool presented in reference [56] is considered suitable for adolescents. Therefore, the authors should clearly prove or provide evidence for this.

Comments on the Quality of English Language

I think the authors have been working very hard to revise this paper, but I am going to suggest comment on a few minor points.

First, the widths of Table 1 and Table 2 still do not match.

Second, information on whether the measurement tools are suitable for university students should be presented in detail. For example,

Line 144-145,

"Impulsivity was measured using the BARRAT-BIS 11 impulsivity scale [55]. In this work, the Spanish adapted version of [56] is used."

The tool presented in reference [56] is considered suitable for adolescents. Therefore, the authors should clearly prove or provide evidence for this.

Author Response

Dear reviewer 1

Thank you very much for the review conducted. All your comments, suggestions, and corrections have been taken into account and applied. Below, we respond to your kind comments.

-- First, the widths of Table 1 and Table 2 still do not match.

We have adjusted the width of tables 1 and 2 to match the rest of the tables.

-Second, information on whether the measurement tools are suitable for university students should be presented in detail. For example,

Line 144-145,

"Impulsivity was measured using the BARRAT-BIS 11 impulsivity scale [55]. In this work, the Spanish adapted version of [56] is used."

 The tool presented in reference [56] is considered suitable for adolescents. Therefore, the authors should clearly prove or provide evidence for this.

We have added the following reference instead of the previous one (Salvo & Castro, 2013). It concerns the Spanish adaptation of the questionnaire in a university population:

 56.Urrego Barbosa, S. C., Valencia Casallas, O. L., & Villalba, J. (2017). Validation of the barrat impulsivity scale (bis-11) in bogotana population. Diversitas: Perspectivas en Psicología, 13(2), 143-157. http://dx.doi.org/10.4067/S0717-92272013000400003

Additionally, we have included the following line in the text: “In this work, the Spanish adapted version of [56] is used, with this adaptation being measured in a university population”. We have also added a brief note in the other instruments indicating that they have been validated in Spanish university population

Thanks in advance for your time. We sincerely hope that with these modifications, we meet your expectations.

We remain attentive to any comments or feedback

Best regards

Itziar Iruarrizaga

Reviewer 2 Report

Comments and Suggestions for Authors

It was confirmed that all suggestions requiring modification were modified.

Author Response

Dear reviewer 2

Thanks for your time. 

Best regards

Itziar Iruarrizaga